# Assessing the Resilience Potential of Inshore and Offshore Coral Communities in the Western Gulf of Thailand

**Makamas Sutthacheep, Charernmee Chamchoy, Sittiporn Pengsakun, Wanlaya Klinthong and Thamasak Yeemin *** 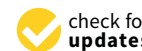

Marine Biodiversity Research Group, Department of Biology, Faculty of Science, Ramkhamhaeng University, Huamark, Bangkok 10240, Thailand; smakamas@hotmail.com (M.S.); charernmee14@hotmail.com (C.C.); marine_ru@hotmail.com (S.P.); klinthong_fai@hotmail.com (W.K.)

**\*** Correspondence: thamasakyeemin@hotmail.com; Tel.: +6623108415

**Abstract:** Coral reefs in the Gulf of Thailand have experienced severe coral bleaching events and anthropogenic disturbances during the last two decades. This study assessed the resilience potential of coral communities at Ko Losin offshore reef sites and Mu Ko Chumphon nearshore coral reefs, in the south of Thailand, by conducting field surveys on the live coral cover, hard substratum composition and diversity and density of juvenile corals. Most study sites had higher percentages of live coral cover compared to dead coral cover. Some inshore and offshore reef sites showed low resilience to coral bleaching events. The total densities of juvenile corals at the study sites were in the range of 0.89–3.73 colonies/m$^2$. The density of the juvenile corals at most reef sites was not dependent on the live coral cover of adult colonies in a reef, particularly for the *Acropora* communities. We suggest that Ko Losin should be established as a marine protected area, and Mu Ko Chumphon National Park should implement its management plans properly to enhance coral recovery and promote marine ecotourism. Other measures, such as shading, should be also applied at some coral reefs during bleaching periods.

**Keywords:** coral; recruitment; resilience; bleaching; management; restoration; fishing; tourism; recovery; Thailand

## 1. Introduction

Coral reefs are recognized as a high-biodiversity ecosystem containing thousands of species that provide socioeconomic benefits. The benefits include providing food and livelihoods for millions of people in tropical countries and the protection of coastal communities from extreme weather disturbances [1,2]. However, coral reefs around the world are degrading because of natural stressors (bleaching, diseases and heavy storms [3–9]) and anthropogenic disturbances, particularly coastal development, pollution, sedimentation and overfishing [10–13]. Human impacts have also reduced the ability of coral recovery and reef resilience after severe disturbances [14–16]. Knowledge about the synergistic effects of coral bleaching and human activities on the ecological processes of coral reefs, particularly coral recruitment, is very important for establishing a science-based management strategy for enhancing the resilience potential of coral reefs [17].

Coral reef management requires supporting ecosystem processes that lower sensitivity, promote recovery, and enhance the adaptive capacity of coral reefs to bleaching by reducing other human impacts [18]. The capacity of coral reefs to resist or recover from degradation and to maintain their ecosystem services is defined as coral reef resilience [19]. Resilience-based management of

coral reefs includes assessing spatial variation in resilience potential and implementing appropriate management plans [18,20,21]. The assessment of the resilience potential of coral reefs was first developed after the coral bleaching event in the year 1998 and it focused on the physical and ecological characteristics of coral reefs that provide some reefs with greater resistance to and/or recovery from coral bleaching [22,23]. Several resilience indicators have been widely developed and proposed for assessing the ecological resilience of coral reefs [24–27].

Successful coral recruitment and juvenile survivorship play an important role in the maintenance of coral populations under normal natural conditions and following mass mortality from bleaching events [28–30]. The planktonic larval stage, settlement and juvenile coral are critical periods in the coral life cycle and have high mortality rates, particularly under stressful environments. Following coral bleaching events, most surviving adult corals show reduced fecundity and growth as well as decreased reproductive outputs and recruitment rates [18,31]. Therefore, coral recruitment is often used as a bioindicator of coral reef health, recovery rate and resilience potential after severe disturbances such as bleaching events. A high coral recruitment rate or high density of juvenile corals on natural substrates can lead to quick coral recovery of degraded reefs after coral bleaching events and anthropogenic disturbances [32]. Coral recovery is also controlled through grazing by herbivores, which limits algal growth [33]. Several environmental factors influence coral recruitment rates, particularly water pollution, overfishing and coastal development, which can affect coral competition ability, fecundity, fertilization success, settlement and survival of juvenile corals [34–36]. Coral recovery and the resilience potential of coral reefs are usually controlled by coral larval supply, recruitment rate, the survival rate of juvenile corals and high resistance/tolerance to environmental stresses [17,37,38].

Mass coral bleaching events in the Gulf of Thailand were reported in 1998, 2010 and 2016 [39–41]. There were significant differences in the susceptibility of coral species to bleaching events in the Gulf of Thailand between the years 1998 and 2010. The 2010 coral bleaching phenomenon at some reef sites, such as Ko Samui in the Western Gulf of Thailand, was more severe than the 1998 bleaching event [39]. The intensive study of coral bleaching in the Gulf of Thailand in the year 2016 revealed that the levels of coral bleaching varied significantly among the reef sites. A high severity level of coral bleaching, of about 70%, was recorded at Ko Ngam Noi, Chumphon Province, in the south of Thailand. The coral mortality following the 2016 bleaching event was approximately 18%, which was much lower than that of the 2010 coral bleaching event because the southwest monsoon started earlier, and therefore the seawater temperature dropped rapidly [41]. Previous studies defined resilience as the capacity of a system to absorb or withstand stressors, maintain its structure and functions in the face of disturbance and change and adapt to future challenges [42,43]. This study aims to assess the resilience potential, based on coverages of live coral, dead coral, rubble and other benthic organisms, of coral communities at Ko Losin offshore reef sites in Pattani Province and Mu Ko Chumphon nearshore coral reefs in Chumphon Province, in the south of Thailand. Field surveys on the live coral cover, hard substratum composition and diversity and density of juvenile corals were conducted to determine the resilience of the coral communities in the south of Thailand.

## 2. Materials and Methods

The study was conducted on coral communities in the Western Gulf of Thailand in March–May 2019. Six study sites from two different groups of coral communities, i.e., three study sites from Ko Losin offshore coral assemblages on pinnacles and three study sites from Mu Ko Chumphon nearshore coral reefs in Mu Ko Chumphon National Park, were selected for this study (Figure 1). Ko Losin is a small isolated island with an old lighthouse giving signals to boat navigators, about 72 km from the mainland. It has a relatively high water clarity in the Gulf of Thailand and harbors coral reefs that are well developed in deeper water, extending from 7 to 25 m depth. Ko Losin has been affected by fishing activities as it is an unprotected remote area. Recently, it is also used as a diving site in the Gulf of Thailand during the southwest monsoon period. Mu Ko Chumphon National Park is a marine protected area that is managed by the Department of National Parks, Plant and Wildlife Conservation. There are

about 40 nearshore islands in Chumphon Province in the Western Gulf of Thailand, which harbor several coral reefs in good condition with high potential for tourism, particularly snorkeling and SCUBA diving. Three reef sites in Mu Ko Chumphon, i.e., Ko Kula, Ko Ngam Yai and Ko Ngam Noi, were selected for the field surveys. The coral reefs at the study sites were in shallow water, 1–12 m in depth. Ko Kula had relatively turbid water as it was affected by high sediment load from the mainland. The location, environmental conditions and anthropogenic disturbances at each study site are summarized in Table 1.

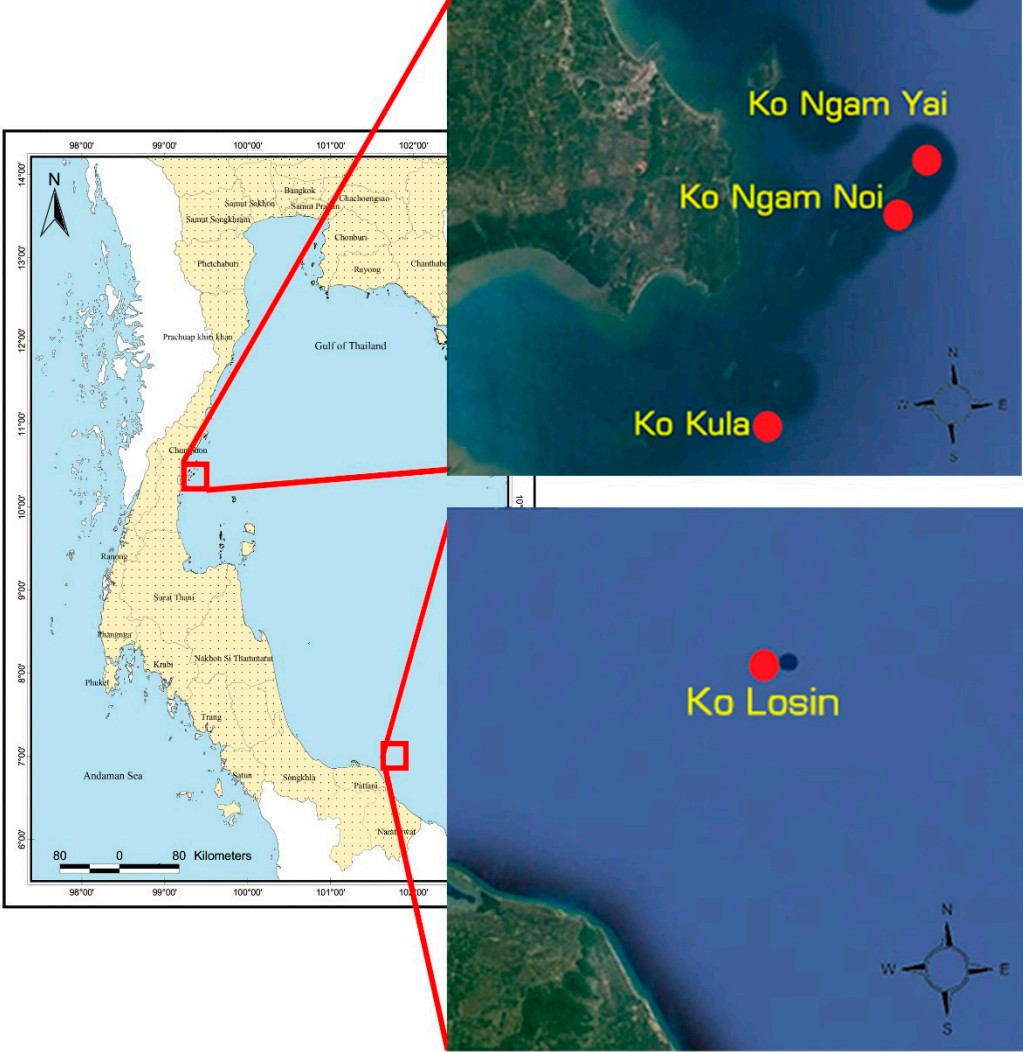

**Figure 1.** Map of the study sites at Mu Ko Chumphon National Park, Chumphon Province and Ko Losin, Pattani Province.

**Table 1.** Location and information of the study sites in the south of Thailand.

| Study Sites | Latitude (N), Longitude (E) | Exposure Condition | Coral Reef Type | Distance from the Shore (km) | Water Transparency | Depth (m) | Anthropogenic Disturbances |
|---|---|---|---|---|---|---|---|
| **Pattani Province** | | | | | | | |
| Ko Losin (West) | 7°19.376′ N 101°53.298′ E | Exposed | Developing reef | 72 | Clear | 8–25 | Tourism (low), Fishery (high) |
| Ko Losin (South) | 7°18.830′ N 101°53.900′ E | Exposed | Developing reef | 72 | Clear | 10–20 | Tourism (low), Fishery (high) |
| Ko Losin (East) | 7°19.484′ N 101°54.340′ E | Exposed | Developing reef | 72 | Clear | 7–20 | Tourism (low), Fishery (high) |
| **Chumphon Province** | | | | | | | |
| Ko Kula | 10°15.347′ N 99°15.205′ E | Sheltered | Fringing reef | 5.5 | Turbid | 1–5 | Tourism (high), Fishery (low) |
| Ko Ngam Yai | 10°29.531′ N 99°25.120′ E | Sheltered | Fringing reef | 21 | Clear | 1–6 | Tourism (high), Fishery (low) |
| Ko Ngam Noi | 10°29.200′ N 99°25.060′ E | Sheltered | Fringing reef | 20.5 | Clear | 1–12 | Tourism (high), Fishery (low) |

At each study site, the live coral cover was observed and evaluated as colony area/unit area in three belt-transects of $50 \times 1$ m$^2$, coral colonies ($\geq 5$ cm in diameter) were counted and identified to the species level [44], if possible, and their coverage was quantitatively estimated. Covers of dead corals, rubble, sand, rock and other benthic components were recorded. In this study, covers of dead corals, rubble, rock and other benthic components were combined as available substrate. Quadrats were also photographed with an underwater camera for reinvestigating the data. Quadrats ($50 \times 50$ cm$^2$ each) were randomly placed on available substrates at each study site by SCUBA divers, and the number of juvenile coral colonies ($\leq 5$ cm in diameter) was carefully observed, identified, counted and photographed for reconfirmation in the laboratory. All juvenile coral colonies were identified to the genus level [44].

Cluster analysis and the non-multidimensional scaling method were performed to categorize study sites on the basis of the Bray–Curtis similarity of benthic components, using PRIMER version 7.0. Differences in the taxonomic composition of corals between Ko Losin and Mu Ko Chumphon were tested by analysis of similarities (ANOSIM), and the coral species contributing most to the dissimilarity between the study sites were identified by similarity percentage (SIMPER) analyses. A one-way ANOVA was used to test the differences in the percentages of live coral cover, species diversity and juvenile coral densities among study sites. Where significant differences were found, the Tukey HSD (honestly significant difference) test was employed to determine which reef site(s) differed.

## 3. Results

There were significant differences in coral cover among study sites (one-way ANOVA, $p < 0.05$) (Figures 2 and 3). The highest percentages of live coral cover were found at Ko Ngam Noi ($77.3 \pm 9.3$) and Ko Kula ($57.7 \pm 6.9$) in Mu Ko Chumphon and at Ko Losin (West) ($47.0 \pm 18.0$), Ko Losin (East) ($45.7 \pm 20.5$) and Ko Losin (South) ($26.7 \pm 10.2$), while the lowest coverage was observed at Ko Ngam Yai ($5.4 \pm 0.6$). All study sites except Ko Ngam Yai had a higher percentage of live coral cover compared to dead coral cover.

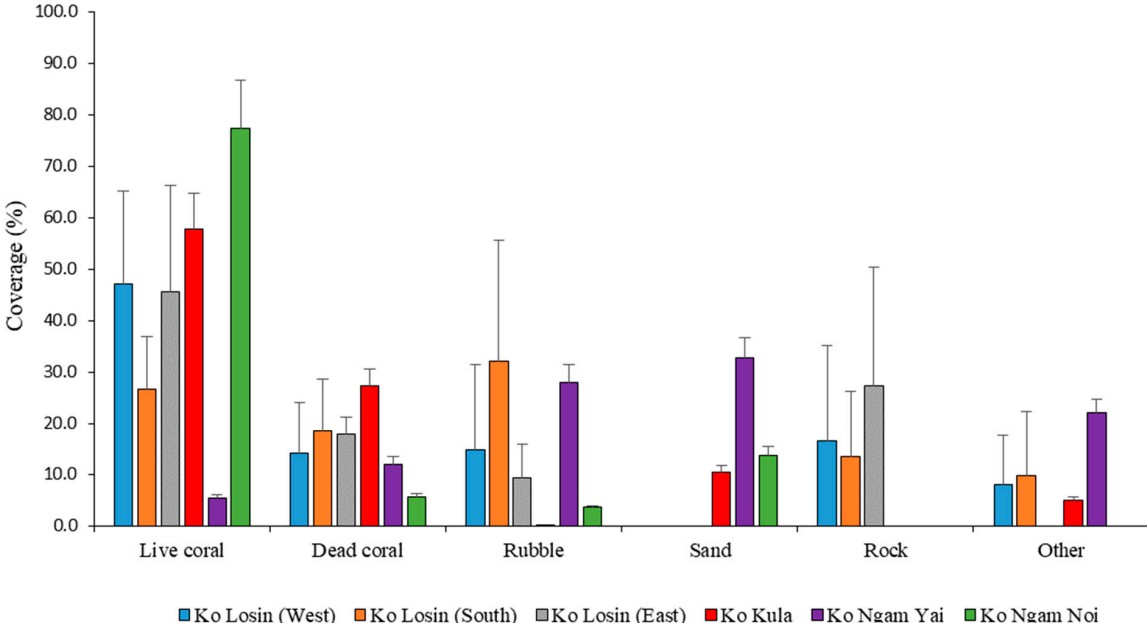

**Figure 2.** Average percentage cover of live corals, dead corals and other benthic components at the study sites. Error bars indicate standard deviation.

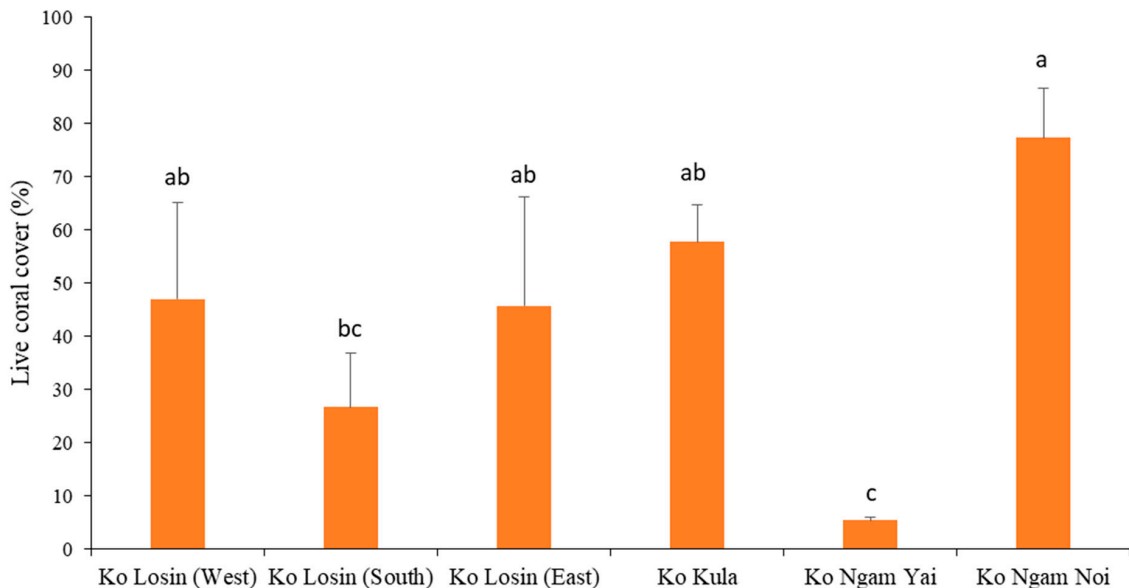

**Figure 3.** Live coral cover at the study sites (one-way ANOVA, $p < 0.05$). Error bars indicate standard deviation. Different letters above bars indicate statistical differences ($p < 0.05$), as determined by Tukey's HSD.

All reef sites except Ko Kula harbored relatively high coral diversity. The highest resilience potential site was Ko Ngam Noi, which was dominated by *Acropora* spp. The high potential sites included Ko Kula, Ko Losin (West) and Ko Losin (East), while the low resilience-potential sites were Ko Ngam Yai and Ko Losin (South), which were dominated by *Porites lutea* (Figure 4). Overall, only Ko Ngam Yai had low resilience potential in terms of survival after bleaching and anthropogenic disturbances. The Shannon–Wiener index of diversity (H′) was significantly different among the six study sites (one-way ANOVA, F = 25.27, $p = 0.001$). Tukey HSD tests showed that Ko Losin (East) was more diverse (H′ = 1.7 ± 0.2) than Ko Kula (H′ = 0.5 ± 0.1) (Figure 5).

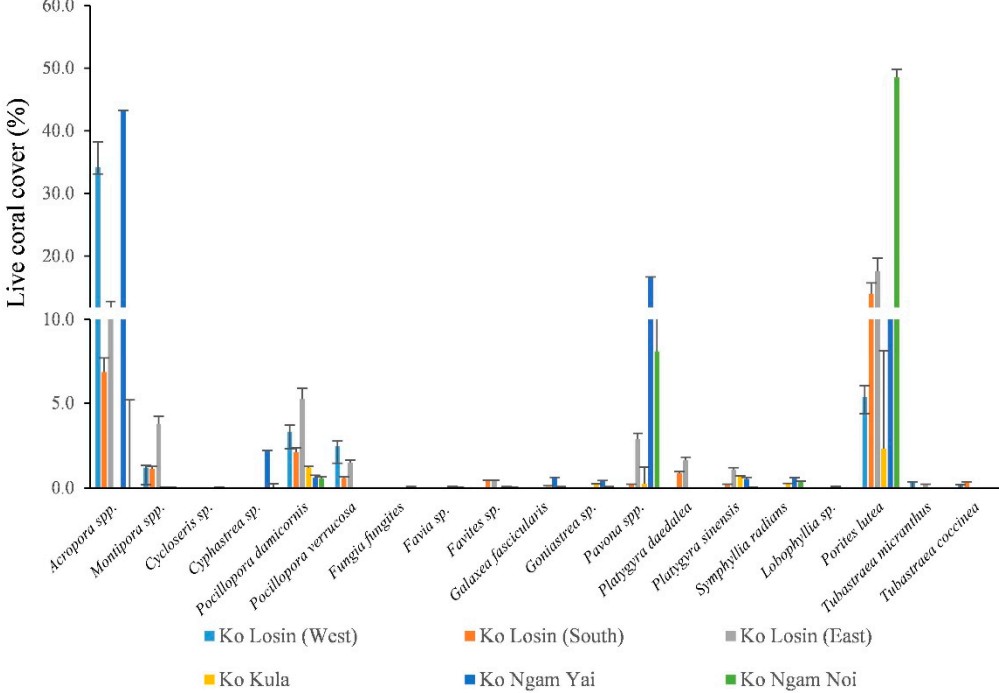

**Figure 4.** Species composition of corals at the study sites. Error bars indicate standard deviation.

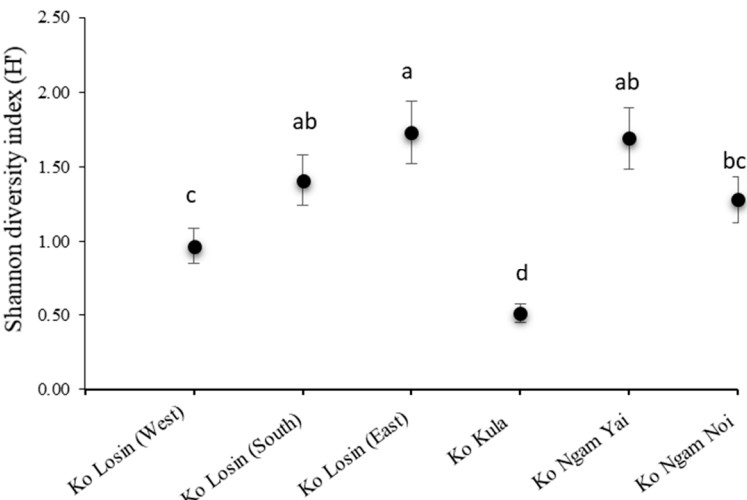

**Figure 5.** Shannon–Wiener index of diversity (mean ± SD) of coral species for each study site (one-way ANOVA, $p < 0.05$). Different letters above indicate statistical differences ($p < 0.05$), as determined by Tukey's HSD.

ANOSIM indicated significant differences in the taxonomic composition of corals between Ko Losin and Mu Ko Chumphon (R = 0.52, $p < 0.001$, Figure 6). The average similarity in the composition of coral species between Ko Losin and Mu Ko Chumphon ranged from about 41.64% to 69.62%, whereas dissimilarity between Ko Losin and Mu Ko Chumphon was 59.74% (SIMPER analysis), Table 2.

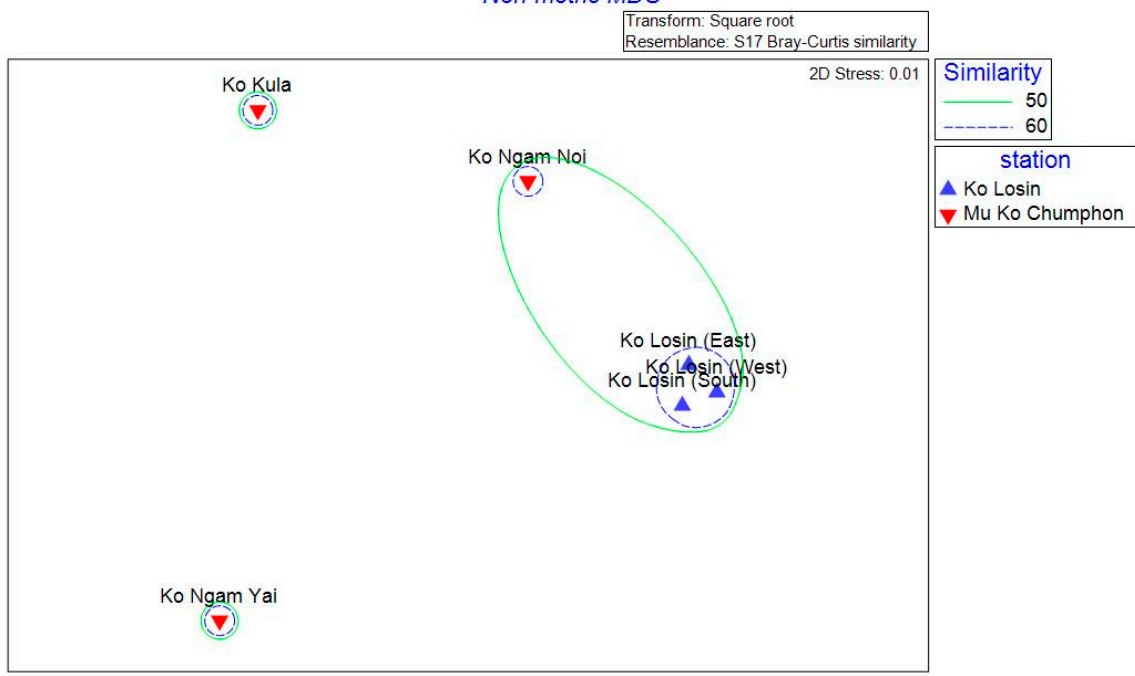

**Figure 6.** Two-dimensional non-metric multidimensional scaling (NMDS) plot of the taxonomic composition of corals at the study sites.

**Table 2.** Similarity percentage (SIMPER) analysis of benthic communities in two regions in the Gulf of Thailand.

| SIMPER | Average Dissimilarity (%) |
|---|---|
| **Ko Losin and Mu Ko Chumphon** | |
| *Acropora* spp. | 13.69 |
| *Porites lutea* | 8.16 |
| *Pavona* spp. | 6.71 |
| *Montipora* spp. | 4.54 |
| *Pocillopora verrucosa* | 4.43 |
| *Pocillopora damicornis* | 3.46 |
| *Platygyra daedalea* | 2.68 |
| *Symphyllia radians* | 2.22 |
| *Platygyra sinensis* | 1.80 |
| *Goniastrea* sp. | 1.59 |
| *Favites* sp. | 1.45 |
| *Cyphastrea* sp. | 1.36 |
| *Tubastraea coccinea* | 1.35 |
| *Galaxea fascicularis* | 1.31 |

The two-dimensional non-metric multidimensional scaling (NMDS) plot of the study sites based on the live corals, dead corals and other benthic components revealed that there were three groups of study sites, i.e., all three study sites of Ko Losin, Ko Kula and Ko Ngam Noi study sites, and Ko Ngam Yai study site (Figure 7).

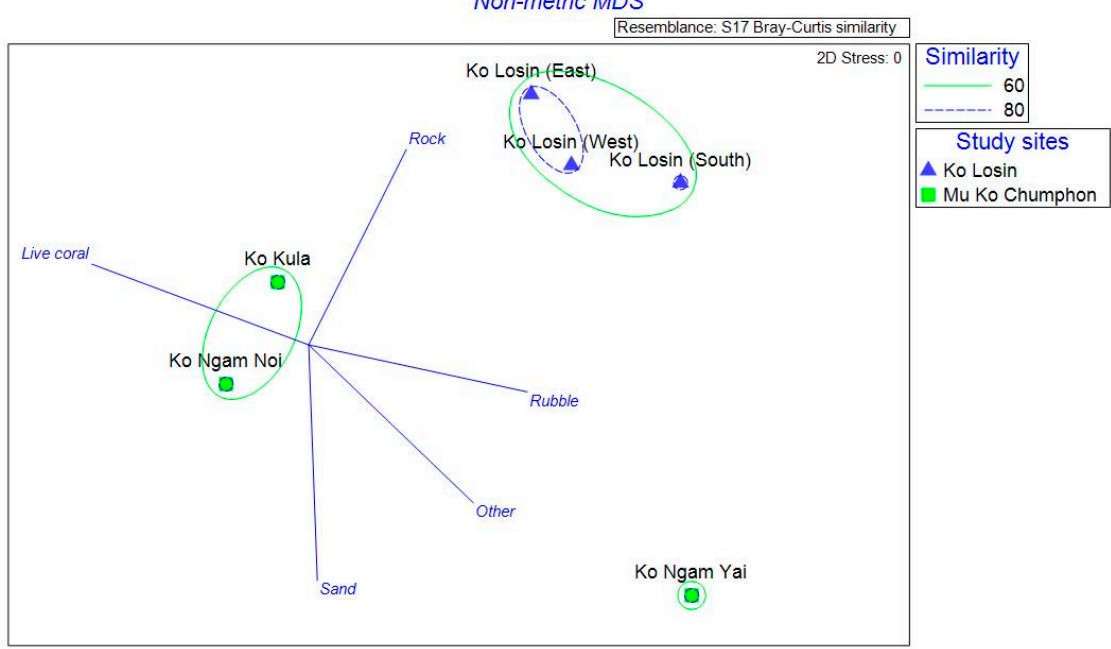

**Figure 7.** Two-dimensional NMDS plot of the study sites.

Underwater photographs of the six study sites are shown in Figure 8. All study sites at Ko Losin and Ko Ngam Noi still displayed high live coral cover of *Acropora* spp., indicating that these reef sites were highly resilient to the coral bleaching events in 1998, 2010 and 2016.

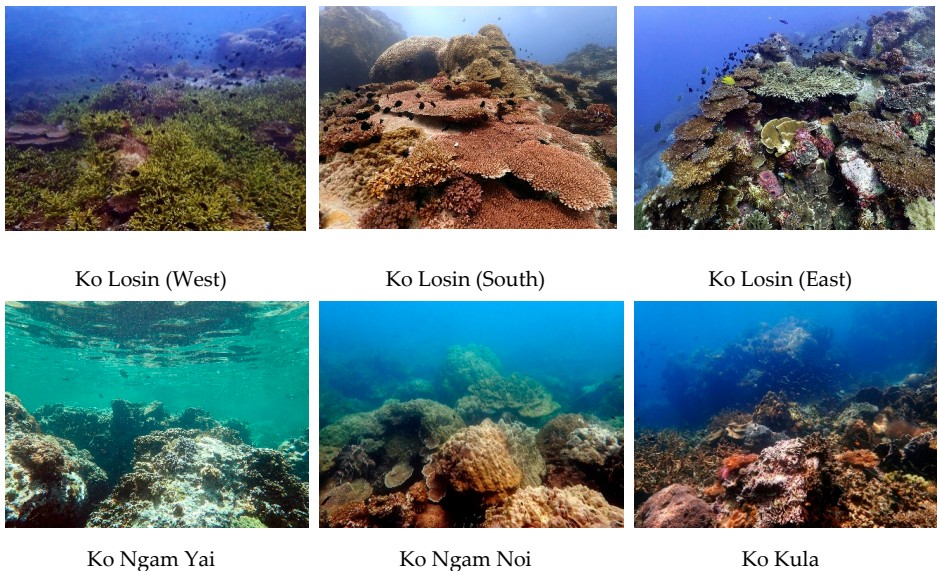

Ko Losin (West)          Ko Losin (South)          Ko Losin (East)

Ko Ngam Yai          Ko Ngam Noi          Ko Kula

**Figure 8.** Underwater photographs showing the dominant coral species at the study sites.

The total densities of juvenile corals, i.e., those less than 5 cm in diameter, at the study sites were in the range of 0.89–3.73 colonies/m$^2$. The highest average density of juvenile corals was found at Ko Ngam Yai (3.73 colonies/m$^2$), while the lowest average density was found at Ko Losin (West) (0.89 colonies/m$^2$). The total density of juvenile corals at Ko Ngam Yai was significantly higher than that at Ko Ngam Noi, Ko Kula and all study sites of Ko Losin (one-way ANOVA; Tukey's HSD test; $p < 0.05$) (Figure 9). A total of 19 genera of juvenile corals were commonly observed, namely, *Pocillopora*, *Tubastrea*, *Montipora*, *Galaxea*, *Pavona*, *Pachyseris*, *Fungia*, *Lithophyllon*, *Hydnophora*, *Turbinaria*, *Lobophyllia*, *Favia*, *Favites*, *Oulastrea*, *Leptastrea*, *Cyphastrea*, *Porites*, *Goniopora* and *Plerogyra*. The juvenile corals of *Pocillopora* were dominant at all study sites except Ko Kula. The most dominant juvenile corals at the study sites of Ko Losin were *Pocillopora*, *Porites* and *Tubastrea*, while the dominant juvenile corals at the study sites of Mu Ko Chumphon were *Pocillopora*, *Porites*, *Fungia*, *Pachyseris*, *Pavona*, *Favites* and *Leptastrea* (Figure 10).

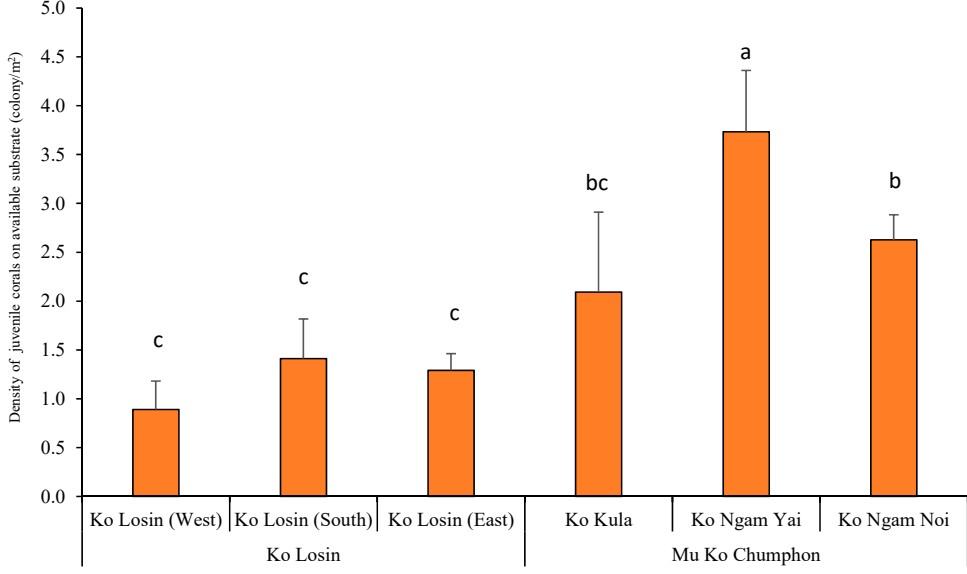

**Figure 9.** Densities of juvenile corals (mean ± SD) on available substrate at the study sites (one-way ANOVA, $p < 0.05$). Different letters above bars indicate statistical differences ($p < 0.05$), as determined by Tukey's HSD.

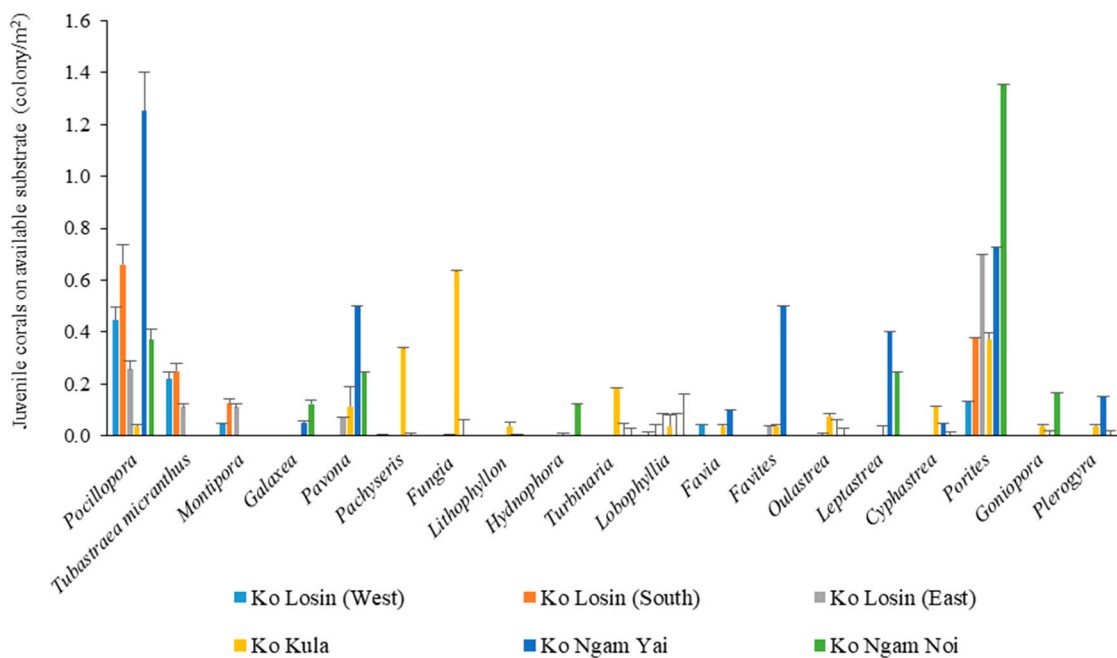

**Figure 10.** Composition of the juvenile corals on available substrate at the study sites. Error bars indicate standard deviation.

ANOSIM indicated significant differences in the composition of juvenile corals between Ko Losin and Mu Ko Chumphon (R = 0.63, *p* < 0.001, Figure 11). The average similarity in the composition of juvenile corals between Ko Losin and Mu Ko Chumphon ranged from about 43.17% to 73.68%, whereas dissimilarity between Ko Losin and Mu Ko Chumphon was 63.81% (SIMPER analysis), Table 3.

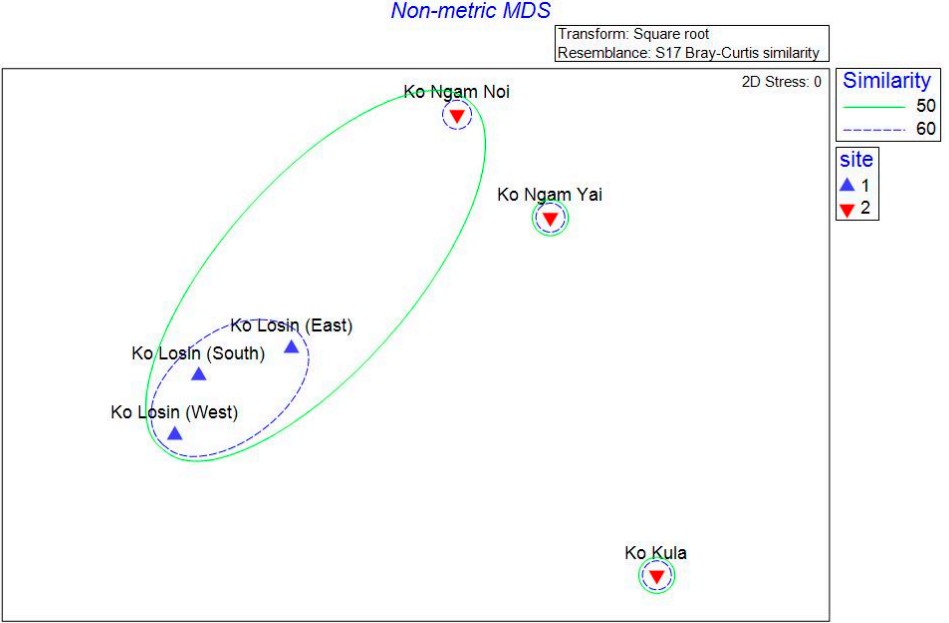

**Figure 11.** Two-dimensional NMDS plot of the composition of juvenile corals at the study sites.

**Table 3.** SIMPER analysis of the composition of juvenile corals at the study sites.

| SIMPER | Average Dissimilarity (%) |
|---|---|
| **Ko Losin and Mu Ko Chumphon** | |
| *Tubastraea micranthus* | 4.57 |
| *Pavona* spp. | 2.06 |
| *Leptastrea* spp. | 1.33 |
| *Porites* spp. | 1.15 |
| *Pocillopora* spp. | 1.79 |
| *Montipora* spp. | 4.60 |
| *Favites* spp. | 0.98 |
| *Fungia* spp. | 0.67 |
| *Goniopora* spp. | 1.09 |
| *Galaxea* spp. | 1.17 |
| *Pachyseris* spp. | 0.67 |
| *Plerogyra* spp. | 1.18 |
| *Cyphastrea* spp. | 1.24 |
| *Favia* spp. | 1.22 |
| *Turbinaria* spp. | 0.67 |

The juvenile coral densities of the brooder *Pocillopora* were relatively high at Ko Ngam Noi ($0.37 \pm 0.15$ colonies/m²), Ko Losin Pinnacle (South) ($0.66 \pm 0.08$ colonies/m²) and Ko Losin (West) ($0.44 \pm 0.05$ colonies/m²). The juvenile coral densities of broadcast spawners at the study sites of Mu Ko Chumphon were much higher compared to those at the study sites of Ko Losin (Figure 12). Underwater photographs of the dominant juvenile corals, *Pocillopora*, *Porites* and *Tubastraea*, at the six study sites are shown in Figure 13. The juvenile corals were in healthy conditions without any signs of partial mortality or stress from competitors, diseases and bleaching.

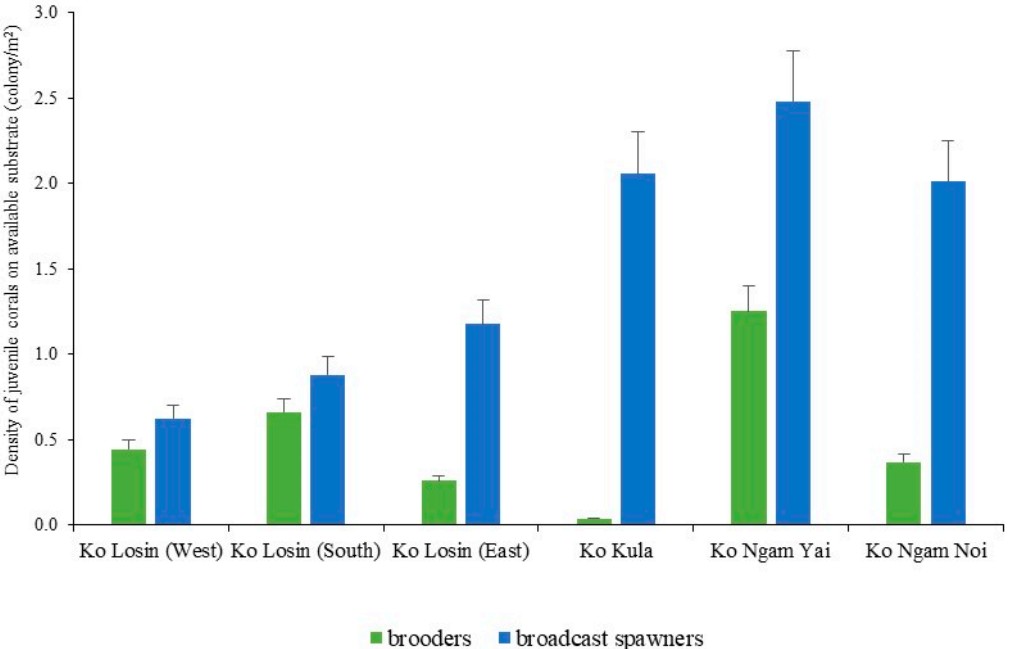

**Figure 12.** Densities of juvenile corals on available substrate for broadcast spawners and brooders at the study sites. Error bars indicate standard deviation.

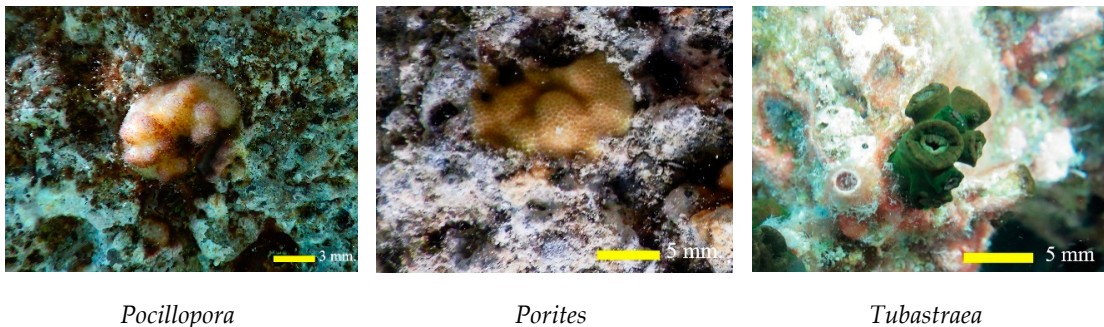

|  |  |  |
|:---:|:---:|:---:|
| *Pocillopora* | *Porites* | *Tubastraea* |

**Figure 13.** Dominant juvenile corals on available substrate at the study sites.

## 4. Discussion

The coral reefs in the Gulf of Thailand are developed in high turbidity and have experienced severe coral bleaching events during the last two decades. The impacts of coastal development, destructive fishing and the expansion of tourism on coral reefs are documented [12,40]. The coral communities at Ko Losin (West), Ko Losin (East) and Ko Ngam Noi are interesting due to their high percentages of live coral cover and the fact that the dominant corals of these reef sites are several species of *Acropora*, which are susceptible to abnormal high-temperature-driven coral bleaching [7,45]. The coral communities at the study sites of Ko Losin are in relatively deep water, which may have protected them from high temperatures during the severe coral bleaching events in 1998 and 2010. Some *Acropora* corals also showed a high degree of bleaching but they did not die after bleaching. Intensive studies on ocean currents and other related issues of physical oceanography are required for understanding high resistance to bleaching events. Protection of the coral communities at Ko Losin from negative impacts of human activities, particularly fishing, boat anchoring and diving, is urgently needed to enhance coral reef resilience in the Gulf of Thailand.

The density of juvenile corals in the Gulf of Thailand is usually lower compared to that of other reef sites in the Indo-Pacific region [46]. Therefore, the coral communities in the Gulf of Thailand can maintain their community structures through the survival of resistant and/or tolerant coral species. The results of this study suggest that highly resistant and tolerant coral species at Ko Losin, Ko Ngam Noi and Ko Kula play a major role in the high resilience potential of coral communities after coral bleaching events. The *Acropora* communities at Ko Ngam Noi, Mu Ko Chumphon National Park, are particularly important to the high resilience potential of nearshore reef sites. These coral communities may provide larval supply to nearshore reefs along the Western Gulf of Thailand through the connecting sea surface current in the Gulf of Thailand [47].

The poor coral condition at Ko Ngam Yai and the high percentage of dead corals at Ko Kula in Mu Ko Chumphon National Park imply the need for urgent investigation on how to restore these reef sites. The densities of juvenile corals at Ko Ngam Yai and Ko Kula from this study were relatively high compared to those of other reef sites in the Gulf of Thailand. The dominant juvenile corals at Ko Ngam Yai were *Pocillopora*, *Porites*, *Favites* and *Pavona*, whereas the dominant juvenile corals at Ko Kula were *Fungia*, *Porites* and *Pachyseris*. Enhancing the survival rates of juvenile corals is crucial for coral recovery following bleaching events [38]. Sediment loaded from coastal development and tourism impacts should be carefully mitigated for passive coral reef restoration. A high diversity of healthy corals in a coral reef ecosystem is an important factor for enhancing reef resilience potential because it occupies the reef substrates and inhibits the settlement of other benthic organisms that are coral competitors [17]. The coral communities at Ko Kula and Ko Ngam Yai also require an adequate supply of coral larvae from other coral reefs in the Gulf of the Thailand to enhance their coral diversity.

The density of juvenile corals recorded in our study was 0.89–3.73 colonies/m$^2$, which is comparable to that of the Palk Bay reef in the northern Indian Ocean [17] but is much lower than that of several reef sites in the Indo-Pacific region, in which the juvenile coral density at some reef sites was over 50 colonies/m$^2$ [48,49]. Variation in the juvenile coral density between the study sites of Mu

Ko Chumphon and Ko Losin was obviously shown in this study. Several factors may influence this spatial variation in juvenile coral density, such as larval supply from the parent reef, larval mortality, reef connectivity, settlement and post-settlement mortality, grazing and sedimentation [50,51]. The density of the juvenile corals at Ko Losin (West), Ko Losin (East), Ko Ngam Noi and Ko Kula was not dependent on the live coral cover of adult coral colonies in a reef. Moreover, the *Acropora* communities at Ko Losin and Ko Ngam Noi had no juvenile corals in their communities.

This study shows that several coral reefs at Ko Losin and Mu Ko Chumphon in the south of Thailand had high resilience potential to coral bleaching events and anthropogenic disturbances because of their survival rates, although they had relatively low densities of juvenile corals. We suggest that Ko Losin should be established as a marine protected area under Thai laws to protect the healthy corals as well as to provide coral larvae to other coral reefs in the Gulf of Thailand. The results from this study also imply that Mu Ko Chumphon National Park should implement its management plans properly to enhance coral recovery at Ko Ngam Yai and Ko Kula. Resilience-based management may be applied to support natural processes that promote the resistance and recovery of corals [43]. The promotion of marine ecotourism can protect coral communities at tourist destinations as well as keep the tourist numbers below the carrying capacity of the reef sites. Other measures to enhance the resistance of corals during bleaching events and appropriate coral restoration projects should be also considered. The field shading experiments that were carried out on coral communities of Ko Ngam Noi should be applied at other reef sites to protect corals during bleaching periods [41].

**Author Contributions:** All of the authors collected data; M.S. and T.Y. conceived the idea; M.S., T.Y., C.C., S.P. and W.K. analysed the data and wrote the manuscript.

**Funding:** This research was funded by Thailand Science Research and Innovation (TSRI), National Science and Technology Development Agency (NSTDA) and a budget for research promotion from the Thai Government to Ramkhamhaeng University.

**Acknowledgments:** We thank Loke Ming Chou and Danwei Huang, the subject editors of this special issue, for encouraging us to analyze the coral reef data from Thailand. We are most grateful to the staff of Marine National Park Operation Center Chumphon, Department of National Park, Wildlife and Plant Conservation, Save Our Sea Association (SOS) and Marine Biodiversity Research Group, Faculty of Science, Ramkhamhaeng University, for their support and assistance in the field. This research was funded by the Thailand Science Research and Innovation (TSRI), National Science and Technology Development Agency (NSTDA) and a budget for research promotion from the Thai Government to Ramkhamhaeng University. We also thank three anonymous reviewers for providing valuable comments and editing the manuscript.

**Conflicts of Interest:** The authors declare no conflict of interest.

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
