# Peer review of "Assessing the Resilience Potential of Inshore and Offshore Coral Communities in the Western Gulf of Thailand"

_jmse, doi:10.3390/jmse7110408_

Round 1
Reviewer 1 Report
This is a descriptive paper on the current status of coral communities at the six coral reef sites in the south western Gulf of Thailand.
Firstly, after reading the paper, I was confused how authors evaluated the resilience potential of coral communities at each reef. They need to adequately describe the definition of resilience potential of coral communities and logic behind it in the Introduction of the paper. Is it current recruit abundance and/or coral cover (definition)? If so, why do you think those variables can be used as indices of resilience potential (logic)?
Secondly, the topic of this paper is most suitable for local Thailand journals or regional journal, not for international journal.
Finally, the manuscript needs to be written more carefully. I could not see Fig. 1. The field shading experiment suddenly appeared at the end of abstract and discussion, but not relevant to the main body of the paper. The methods need to be described in more details.
Author Response
Response to Reviewer 1 Comments
Point 1:
Firstly, after reading the paper, I was confused how authors evaluated the resilience potential of coral communities at each reef. They need to adequately describe the definition of resilience potential of coral communities and logic behind it in the Introduction of the paper. Is it current recruit abundance and/or coral cover (definition)? If so, why do you think those variables can be used as indices of resilience potential (logic)?
Response 1: Thank you very much. We found your comments helpful and have revised accordingly.
Line 84-89 “Previous studies defined resilience as capacity of a system to absorb or withstand stressors and the system can maintain its structure and functions in the face of disturbance and change, and the capacity to adapt to future challenges [42,43]. This study aims to assess resilience potential, based on coverages of live coral, dead coral, rubble and other benthic organisms, of coral communities at Ko Losin, offshore reef sites in Pattani Province and Mu Ko Chumphon, nearshore coral reefs in Chumphon Province, the south of Thailand.”
Point 2:
Secondly, the topic of this paper is most suitable for local Thailand journals or regional journal, not for international journal.
Response 2: The topic of this paper is changed to be “Assessing resilience potential of inshore and offshore coral communities in the Western Gulf of Thailand”
Point 3:
Finally, the manuscript needs to be written more carefully. I could not see Fig. 1. The field shading experiment suddenly appeared at the end of abstract and discussion, but not relevant to the main body of the paper. The methods need to be described in more details.
Response 3:
Figure 1. Map of study sites at Mu Ko Chumphon, Mu Ko Chumphon National Park and Ko Losin, Pattani Province is added. The abstract and discussion are revised.
- Line 23-24 “Other measures such as shading, should be also
applied at some coral reefs during bleaching period.”
- Line 250-252 “Other measures to enhance resistance of corals
during bleaching events should be also considered. The field shading
experiments which were carried out on coral communities of Ko
Ngam Noi should be applied at other reef sites to protect corals
during bleaching periods [41].”
The methods are described in more details.
- We add a reference [44] (Line 117-118).
- Line 122-125 “To examine juvenile coral densities, quadrats
(50x50 cm2 for each) were randomly placed on available substrates
at each study site using SCUBA diving and number of juvenile coral
colonies (≤ 5 cm in diameter) was carefully observed, identified,
counted and photographed for reconfirmation in a laboratory and
counted.”

Reviewer 2 Report
This is an important and interesting study, but there needs to be clarification of the methods and results. I do not feel that the results support the overarching statements that the authors claim, such as the reefs are resilient. Perhaps this can be fixed by providing more information. The introduction and discussion should also be expanded to include the other aspects that would affect the resilience or a coral reef.
There are several grammatical errors and overuse of the word "the."

Author Response
Response to Reviewer 2 Comments
This is an important and interesting study, but there needs to be clarification of the methods and results. I do not feel that the results support the overarching statements that the authors claim, such as the reefs are resilient. Perhaps this can be fixed by providing more information. The introduction and discussion should also be expanded to include the other aspects that would affect the resilience or a coral reef.
There are several grammatical errors and overuse of the word "the."
Response: Thank you very much. We found your comments helpful and have revised accordingly as the following points.
Commented [kw1]: The abstract is a bit hard to follow with the site names that have not been introduced. Perhaps this can be clarified/simplified here.
Response1: Thank you very much. We found your comments helpful and have revised accordingly.
Commented [kw2]: This is not the right word for this.
Response2: We have changed some words.
Line 49-51 “Coral reef management requires supporting the supporting ecosystem processes that lower sensitivity, promote recovery and enhance adaptive capacity of coral reefs to bleaching by reducing other human impacts.”
Commented [kw3]: This sentence does not make sense to me.
Response3: We have revised as your suggestion.
Line 60-62 “The planktonic larval stage, settlement and juvenile coral and early recruitment are critical periods in the coral life cycle and have high mortality rates, particularly under stressful environment.”
Commented [kw4]: Check grammar throughout. “the” is used too often and incorrectly.
Response4: We have revised some incorrect grammar. The manuscript will be submitted to MDPI's English editing service.
Commented [kw5]: This is true, but there are several other measures used that are worth mentioning here.
Response5: We have revised as your suggestion.
Line 68-69 “Coral recovery is also controlled through grazing by herbivores limits algal growth [33].”
Commented [kw6]: Run-on sentence.
Response6: We have revised.
Line 82-84 “The coral mortality following the 2016 bleaching event was much lower than the 2010 coral bleaching event because the southwest monsoon started earlier, therefore, the seawater temperature dropped rapidly [41].”
Commented [kw7]: A map of study sites would be very nice. I do not see one here, although the text suggests there should be one.
Response7: Figure 1. Map of study sites at Mu Ko Chumphon, Mu Ko Chumphon National Park and Ko Losin, Pattani Province is added.
Commented [kw8]: What sources were used to ID corals?
Response8: We add a reference [44] (Line 117-118).
Commented [kw9]: It is unclear whether IDs were made in situ or based on photographs. Please clarify the methods.
Response9: We have revised as your suggestion.
Line 122-125 “To examine juvenile coral densities, quadrats (50x50 cm2 for each) were randomly placed on available substrates at each study site using SCUBA diving and number of juvenile coral colonies (≤ 5 cm in diameter) was carefully observed, identified, counted and photographed for reconfirmation in a laboratory and counted.”
Commented [kw10]: Please provide indices used to decide which species were resistant and not resistant
Response10: We have revised as your suggestion.
Line 137-138 “Overall, only Ko Ngam Yai was low resilience potential in terms of survival after bleaching and anthropogenic disturbances.”
Commented [kw11]: This figure needs to be improved. I do not see where live coral, dead coral, and other benthic components are on this image. I only see a differentiation in study site.
Response11: Figure 4 has been revised as your suggestion.
Commented [kw12]: Nice photos!
Commented [kw13]: How did you decide whether they were juvenile or adult corals?
Response13: We have revised as your suggestion.
Line 163-164 “The total densities of juvenile corals, i.e. less than 5 cm in diameter, at the study sites were in a range of 0.89 - 3.73 colonies/m2.”
Commented [kw14]: Italicize genera names throughout manuscript
Response14: We have revised throughout manuscript as your suggestion.
Commented [kw15]: Coral recruitment or number of juvenile colonies?
Response15: We have corrected it as “juvenile colonies”.
Commented [kw16]: Juveniles?
Response16: We have corrected it as “juvenile”.
Commented [kw17]: You have not supported this statement just based on counting juvenile colonies. Please expand on your methods to explain how you can support this statement.
Response17: We have revised as your suggestion.
Line 204 – 207 “The high resilience potential of coral communities at Ko Losin (West) , Ko Losin (East) and Ko Ngam Noi revealed in this study are interesting as their high percentages of live coral cover and the dominant corals of these reef sites were several species of Acropora which are susceptible to anomaly high-temperature driven coral bleaching [7, 46].”
Commented [kw18]: How long were the bleaching events? Perhaps this had something to do with why the corals did not die.
Response18: We have revised as your suggestion.
Line 207 – 209 “The coral communities at the study sites of Ko Losin are in relatively deep water that may protect them from high temperature during the severe coral bleaching events in 1998 and 2010.”
Commented [kw19]: Is there evidence for this?
Response19: We have added a reference.
Line 216 – 217 “These coral communities may provide larval supply to nearshore reefs along the Western Gulf of Thailand as connectivity of sea surface current in the Gulf of Thailand [47].
Commented [kw20]: I don’t think this is shown based on what was presented.
Response20: We have revised as your suggestion.
Line 240 – 242 “This study shows that several coral reefs at Ko Losin and Mu Ko Chumphon in the south of Thailand were high resilience potential to coral bleaching events and anthropogenic disturbances because of their survival rates, although they had relatively low densities of juvenile corals.”

Reviewer 3 Report
Manuscript ID: JMSE-566787
Journal of Marine Science and Engineering
Manuscript Title: Assessing resilience potential of inshore and offshore coral communities in the Western Gulf of Thailand 

General comments:
This paper describes the resilience potential of 6 sites in the Gulf of Thailand and the information is important for local management of coral reefs. The recommendations are feasible and I would encourage the authors to provide more details to improve the clarity of the paper. After the suggestions have been incorporated, the paper should be ready for publication.
More specific comments are as follows (note: the line numbers indicated below follows the PDF, that were generated by the journal).
Introduction
L43. Remove spacing before “bleaching”
L66. Remove spacing before “High”
L82-84. Provide an estimate of the percentage bleaching and mortality rates at Ko Ngam Noi to demonstrate the severity of the 2016 bleaching.
Methods
Table 1. If possible, please provide similar categorization (low/medium/high) for anthropogenic disturbances for all activities (e.g. tourism and fishery) at all sites.
L117. Describe how coral cover was calculated in the belt transect. Did the authors use number of colonies/unit area or colony area/unit area?
L118. As many of the taxonomic names for coral species have been revised, please insert appropriate references for the identification.
L121-122. Remove the sentence “to examine… the data”.
L126. Insert reference used to identify the juvenile corals.
L127-131. The analysis will be more robust if the indices for coral diversity, cover etc are included. More details will be included in the sections below.
Results
L133. Include S.D. along with the values of the mean coral cover for each site.
L135. The coral cover can be compared using statistics to test for differences. Include the analysis in fig 2 as well. The authors can also describe other benthic categories in detail and this is especially important for dead coral cover in Ko Ngam Yai.
L137. The remarks on the low resilience potential can be included after all the indicators have been described.
Figure 2. The Error bars should be S.D and not S.E since the data is obtained from the belt transect. Please check.
L142. Please include indices for coral diversity (e.g. Shannon’s Diversity) and test for differences statistically. The authors will need to describe the composition in detail (mean +/- SD) since the bar charts in Fig 3 does not convey this information. The resilience potential for each site should be concluded after all the indicators have been described (see for example Maynard, J. A., McKagan, S., Raymundo, L., Johnson, S., Ahmadia, G. N., Johnston, L., ... & Van Hooidonk, R. (2015). Assessing relative resilience potential of coral reefs to inform management. Biological Conservation, 192, 109-119.)
Figure 3.The authors can consider doing a nMDS with ANOSIM and SIMPER to test for differences among sites for the taxonomic composition of corals. This will make the analysis more robust.
L150. This section describing the benthic composition should be placed right after figure 2 and before the description of coral community for better flow. The ANOSIM data for the nMDS should be included to establish the differences among different clusters. The SIMPER analysis would be able to give a clearer description of the factors that contributed to the differences and it should be included here.
L157 to L160. This paragraph isn’t very informative. I would suggest removing it and retain figure 5.
L164-165. Include the mean density for all sites and their respective SD. The authors will also need to describe the composition in detail (mean +/- SD) since the bar charts in Fig 7 does not convey this information.
Figure 6. The caption has to describe what the letters and error bars represent.
L184. Keep the names of the sites consistent. For example it should be “Ko Lisin (South)” instead of “Losin Pinnacle (South). Please check throughout the manuscript.
L185. Provide the numerical values for the densities for each site.
Figure 7. The authors can consider doing a nMDS with ANOSIM and SIMPER to test for differences among sites for the taxonomic composition of corals. This will make the analysis more robust.
Discussion
L197-198. These 2 sentences seem out of place. Please revise it to make it flow with the previous sentence. I am confused what the authors are trying to convey here so I can’t provide more specific suggestions.
L207. Please change to “abnormally”
Overall structure. This section requires some reorganization. The authors should discuss the findings for benthic cover, species composition, juvenile coral density and composition in this sequence. In each section, the authors can consider describing how the factors are important for resilience based on previous study (either done in Thailand or elsewhere). This is particular important in this study as some species are known to be more resilient that the others and should be described in more detail. Collectively, all the information will provide the basis for concluding the relative resilience potential among sites and the recommendations will be more convincing.
L250. The authors mentioned in earlier sections that restoration is one possible measure. This is worth reiterating here.
Author Response
Introduction
L43. Remove spacing before “bleaching”
Response: Corrected.
L66. Remove spacing before “High”
Response: Corrected.
L82-84. Provide an estimate of the percentage bleaching and mortality rates at Ko Ngam Noi to demonstrate the severity of the 2016 bleaching.
Response: Provided additional data.
Methods
Table 1. If possible, please provide similar categorization (low/medium/high) for anthropogenic disturbances for all activities (e.g. tourism and fishery) at all sites.
Response: Provided additional data.
L117. Describe how coral cover was calculated in the belt transect. Did the authors use number of colonies/unit area or colony area/unit area?
Response: Provided additional data.
L118. As many of the taxonomic names for coral species have been revised, please insert appropriate references for the identification.
Response: Provided additional data.
L121-122. Remove the sentence “to examine… the data”.
Response: Corrected.
L126. Insert reference used to identify the juvenile corals.
Response: Provided additional data.
L127-131. The analysis will be more robust if the indices for coral diversity, cover etc are included. More details will be included in the sections below.
Response: Provided additional data analyses.
Results
L133. Include S.D. along with the values of the mean coral cover for each site.
Response: Corrected.
L135. The coral cover can be compared using statistics to test for differences. Include the analysis in fig 2 as well. The authors can also describe other benthic categories in detail and this is especially important for dead coral cover in Ko Ngam Yai.
Response: Provided additional data analyses.
L137. The remarks on the low resilience potential can be included after all the indicators have been described.
Response: Corrected.
Figure 2. The Error bars should be S.D and not S.E since the data is obtained from the belt transect. Please check.
Response: Corrected.
L142. Please include indices for coral diversity (e.g. Shannon’s Diversity) and test for differences statistically. The authors will need to describe the composition in detail (mean +/- SD) since the bar charts in Fig 3 does not convey this information. The resilience potential for each site should be concluded after all the indicators have been described (see for example Maynard, J. A., McKagan, S., Raymundo, L., Johnson, S., Ahmadia, G. N., Johnston, L., ... & Van Hooidonk, R. (2015). Assessing relative resilience potential of coral reefs to inform management. Biological Conservation, 192, 109-119.)
Response: Provided additional data analyses.
Figure 3.The authors can consider doing a nMDS with ANOSIM and SIMPER to test for differences among sites for the taxonomic composition of corals. This will make the analysis more robust.
Response: Provided additional data analyses and added figure and table.
L150. This section describing the benthic composition should be placed right after figure 2 and before the description of coral community for better flow. The ANOSIM data for the nMDS should be included to establish the differences among different clusters. The SIMPER analysis would be able to give a clearer description of the factors that contributed to the differences and it should be included here.
Response: Provided additional data analyses and added figure and table.
L157 to L160. This paragraph isn’t very informative. I would suggest removing it and retain figure 5.
Response: Retain figure 5.
L164-165. Include the mean density for all sites and their respective SD. The authors will also need to describe the composition in detail (mean +/- SD) since the bar charts in Fig 7 does not convey this information.
Response: Corrected, revised and added figures.
Figure 6. The caption has to describe what the letters and error bars represent.
Response: Corrected.
L184. Keep the names of the sites consistent. For example it should be “Ko Lisin (South)” instead of “Losin Pinnacle (South). Please check throughout the manuscript.
Response: Corrected.
L185. Provide the numerical values for the densities for each site.
Response: Provided additional data.
Figure 7. The authors can consider doing a nMDS with ANOSIM and SIMPER to test for differences among sites for the taxonomic composition of corals. This will make the analysis more robust.
Response: Provided additional data analyses and added figure and table.
Discussion
L197-198. These 2 sentences seem out of place. Please revise it to make it flow with the previous sentence. I am confused what the authors are trying to convey here so I can’t provide more specific suggestions.
Response: Corrected.
L207. Please change to “abnormally”
Response: Corrected.
Overall structure. This section requires some reorganization. The authors should discuss the findings for benthic cover, species composition, juvenile coral density and composition in this sequence. In each section, the authors can consider describing how the factors are important for resilience based on previous study (either done in Thailand or elsewhere). This is particular important in this study as some species are known to be more resilient that the others and should be described in more detail. Collectively, all the information will provide the basis for concluding the relative resilience potential among sites and the recommendations will be more convincing.
Response: Revised.
L250. The authors mentioned in earlier sections that restoration is one possible measure. This is worth reiterating here.
Response: Corrected.

Round 2
Reviewer 1 Report
I see much improvement in the revised manuscript and I agree that the manuscript is now ready as a descriptive study on those coral reefs in the gulf of Thailand.
However, I still cannot agree with your logic; i.e. current state of coral assemblage indicates “coral resilience potential”. I suspect that you assume the current status of those examined coral assembles are the result of recovery from the previous bleaching events in 2010 and 2016. If so, you need to provide such data (e.g., data on deteriorated coral status just after those bleaching events) and then discuss coral resilience potential based on the various recovery data by comparing data between then and now.
In other cases, you can focus on various coral resilience index such as habitat complexity, water depth, coral juvenile density, herbivore abundance, algal abundance, and etc. However, for such an approach, the current data-set of the present study is small.
Therefore, in my opinion, we cannot evaluate coral resilience potential by using coral cover data. For the detailed reasoning, please refer to Hughes et al. (2010) TREE. I’m sorry that I cannot be positive about your revised manuscript.
Hughes TP, Graham NA, Jackson JB, et al. 2010. Rising to the challenge of sustaining coral reef resilience Trends Ecol Evol 25: 633-42.
Author Response
Response: Thank you very much for your valuable suggestions.
In this paper, we would like to highlight that some coral reefs in the Gulf of Thailand experienced severe coral bleaching events during the last two decades, particularly in 1998 and 2010 as well as the impacts of coastal development, destructive fishing and expansion of tourism. The juvenile coral densities were also relatively low. However, the coral communities still have high percentages of live coral cover and the dominant corals of these reef sites are big healthy colonies of several species of Acropora which are susceptible to abnormally high-temperature driven coral bleaching. Therefore, these coral communities in the Gulf of Thailand can maintain their community structures through the survival of resistant and/or tolerant coral species. Our results suggest that high resistant and tolerant coral species at some study sites play a major role on high resilience potential of coral communities after the coral bleaching events.
